# Effect of Different Biological and Organic Fertilizer Sources on the Quantitative and Qualitative Traits of *Cephalaria syriaca*

**Amir Rahimi** [1], **Reza Amirnia** [1,*], **Sina Siavash Moghaddam** [1], **Hesham Ali El Enshasy** [2,3,4], **Siti Zulaiha Hanapi** [2] and **R. Z. Sayyed** [5]

1   Department of Plant Production and Genetics, Faculty of Agriculture, Urmia University, Urmia 5756151818, Iran; e.rahimi@urmia.ac.ir (A.R.); ss.moghaddam@urmia.ac.ir (S.S.M.)
2   Institute of Bioproduct Development (IBD), Universiti Teknologi Malaysia (UTM), Johor Bahru 81310, Malaysia; henshasy@ibd.utm.my (H.A.E.E.); zulaiha@ibd.utm.my (S.Z.H.)
3   School of Chemical and Energy Engineering, Universiti Teknologi Malaysia (UTM), Johor Bahru 81310, Malaysia
4   City of Scientific Research and Technology Applications (SRTA), New Borg Al Arab 21934, Egypt
5   Department of Microbiology, PSGVP Mandal's Arts, Science, and Commerce College, Shahada 425409, India; sayyedrz@gmail.com
*   Correspondence: r.amirnia@urmia.ac.ir or ramirnia@gmail.com

**Abstract:** Due to the potential to enhance soil productivity and plant growth, biological fertilizers have recently been considered an alternative source for soil, water, and crop-contaminating chemical fertilizers in sustainable agriculture. The importance of different fertilizer sources on quantitative and qualitative traits of *Syrian cephalaria* (*Cephalaria syriaca* L.) was explored in an experiment based on a randomized complete block design during the 2015–2016 growing season. The maximum grain yield (9.97 g/plant) and biological yield (24.57 g/plant) were obtained from the application of *Azotobacter* + chemical fertilizer treatment, but the maximum oil percentage (25.23%) and oil yield (2.41 g/plant) were observed in the plants treated with *Azotobacter* + vermicompost. The plants treated with *Azotobacter* + chemical fertilizer exhibited the highest 1000 seed weight (15.03 g). Application of *Azotobacter* + vermicompost increased chlorophylls *a*, *b*, and total by 2.06, 1.96, and 4.02% versus *Azotobacter* treatment alone, respectively. The treatment of *Azotobacter* + manure increased total phenol, flavonoids, and DPPH antioxidant activity by 27.89, 0.56, and 53.16% versus the treatment of *Azotobacter* + chemical fertilizer. The integrated application of different fertilizer sources had an optimal effect on the uptake of trace elements (Cu, Fe, and Zn) so that the treatment of *Azotobacter* + vermicompost increased their concentrations. Due to the positive effect of integrated application of different fertilizer sources on improving the studied traits of *Cephalaria*, it is recommended to replace chemical fertilizers with combined fertilizers including organic and biological inputs to enhance the efficiency of crops, reduce environmental pollution, and move towards sustainable agriculture.

**Keywords:** antioxidant activity; biological fertilizers; *Cephalaria syriaca* L.; sustainable agriculture; trace elements

## 1. Introduction

Syrian cephalaria (*Cephalaria syriaca* L.) is an annual plant species that grow in natural conditions, widely distributed in some regions, such as Turkey and some European nations, including the south of France and the south of Spain, as well as the African continent. In addition to its chilling resistance, *Cephalaria* can be grown in poor lands. It has shallow roots (60–120 cm) with robust stems that grow vertically. Hairs 4–5 mm long cover the stems and leaves. The plant has a high stemming potential. Since its reproductive organs are located at the end of the main stem and auxiliary branches, there is a direct relationship between seed yield and branch number [1]. Several bioactive compounds, such as triterpenes, flavonoids, glycosides, and alkaloids, were found in the species, which qualify them to be

extensively used in the pharmaceutical industries [2]. Meanwhile, *Cephalaria* has a good oil content (21–26%), so it is sometimes considered as an oilseed, but the oil of this plant has a more industrial aspect due to the presence of epoxy acid [3,4]. The oil extracted from its seeds has a pleasant scent and is greenish-yellow in color.

The application of organic and biological fertilizers has gained growing importance given the devastating effects of chemical fertilizers on the environment [5]. It is impossible to achieve sustainable and organic farming without considering soil biodiversity. Most soil microorganisms play a critical role in converting organic into inorganic matters, supplying nutrient requirements of different plant species and increasing soil fertility by biologically fixing nitrogen and converting some nutrients from unavailable forms into the plan available forms [6]. Azotobacters can produce antifungal compounds to control plant diseases and strengthen germination and seedling vigor, improving plant growth [7].

Biological fertilizers are not merely referred to as organic matters derived from manures, plant residues, etc., but they encompass the products from the activity of microorganisms concerning nitrogen fixation or phosphorus availability [8]. Various microorganisms such as *Azotobacter, Azospirillum, Pseudomonas*, and *Bacillus* play a significant role in the secretion of plant hormones [9]. Vessey [10] reported that the plant growth improvement induced by inoculation with *Azotobacter* and *Azospirillum* was due to N fixation or the synthesis of growth-stimulating hormones, such as cytokinin, gibberellic acid, auxin, amino acids, and B-group vitamins. As one of the organic fertilizer sources, vermicompost is a microbiologically active organic compound with high loading of macro and micronutrients and is produced from the interaction of soil worms and microorganisms during the decomposition of organic matter [11]. It is applied in sustainable agriculture to improve soil porosity and thus nutrient availability by releasing some organic acids, such as oxalic acid. Vermicompost is rich in growth hormones and vitamins, so it increases the microbial population of soils and contributes to the long-term retention of nutrients without adversely affecting the environment [12]. Manures are organic sources of nutrients for sustainable crop production, which supply nutrients and increase the organic matter content of soils, enhancing nutrient absorbability by plants and maintaining a relative balance of nitrogen. They increase seed germination and root and stem growth and development [13].

Since *Syrian cephalaria* (*Cephalaria syriaca*) was studied in Iran for the first time, its production is recommended mainly for its oil and as an alternative for chemical preservatives in the food industry. On the other hand, it is essential to produce valuable species such as *Cephalaria* to note that crop production should be increased without applying harmful chemical inputs. Thus, the present research aimed to explore the effect of different fertilizer sources on the quantitative and qualitative traits of *Syrian cephalaria* grown in Iran.

## 2. Materials and Methods

The study was carried out in a randomized complete block design with six treatments and three replications in the research farm of the Agriculture Department of Urmia University, Urmia, Iran, during 2015–2016. The experimental treatments are including (1) control (no fertilization), (2) *Azotobacter*, (3) *Azotobacter* + manure, (4) *Azotobacter* + vermicompost, (5) *Azotobacter* + phosphate Barvar II, and (6) *Azotobacter* + chemical fertilizer. The fertilizer treatments were applied to the plots as per the research design map.

The seeds of *Cephalaria* were supplied from the Food Industries Group of Ankara University, Turkey. Before sowing, the seeds were inoculated according to the procedure recommendation of Green Biotech Inc., with biological *Azotobacter* fertilizer and Phosphate Barvar II, which contained *Azotobacter vinelandii* (the strain O4), strain P5 of Pantoea, and strain P13 of *Pseudomonas putida* (concentration formulated product: 109 CFU/g). The biofertilizer was mixed with water and was sprayed on the seeds to create a uniform cover over them. Then, the seeds were dried in shade condition before sowing. Based on the soil properties, fertilizers were applied to the soil as follows: vermicompost (6.8 ton ha$^{-1}$), manure (6.3 ton ha$^{-1}$), and chemical fertilizer (Urea: 110 kg ha$^{-1}$ + Triple superphosphate: 60 kg ha$^{-1}$ + Potassium sulfate: 50 kg ha$^{-1}$ + Micronutrients: 23 kg ha$^{-1}$). It should be

noted that half the nitrogen and phosphorus chemical fertilizers were used in treatments that included biofertilizers. The rows were spaced by 25 cm, and the on-row spacing was set at 10 cm. The seeds were sown in mid-March. They had a vigor of 94%. Before the experiment, the soil was sampled (0–30 cm) to determine its physical and chemical characteristics, whose results are presented in Table 1.

**Table 1.** Some physicochemical characteristics of the soil in the studied site.

| EC (dS/m) | pH | Texture | Clay | Silt | Sand | CCE [a] | SP | N | OC [b] | Mn | B | Zn | Fe | K | P |
|---|---|---|---|---|---|---|---|---|---|---|---|---|---|---|---|
| | | | | | | (%) | | | | | | (mg/kg) | | | |
| 1.37 | 7.81 | Clay loam | 43 | 35 | 22 | 15.83 | 55 | 0.06 | 1.18 | 11.5 | 0.3 | 1 | 9.1 | 295 | 9.1 |

[a] Carbonate calcium equivalent. [b] Organic carbon.

All agronomic operations were applied to all treatments uniformly. After full maturity (late July), the experimental plots were harvested separately to record their yield and components. At the physiological maturity stage, the agronomic traits of seed yield, biological yield, harvest index, and 1000 seed weight were measured on 10 plants per plot. Samples were oven-dried at 70 °C for 24 h for phytochemical analyses. The chlorophyll *a*, *b* and total were expressed as follows [14].

$$\text{Chlorophyll } a = 11.24 \times A_{662} - 2.04 \times A_{645} \tag{1}$$

$$\text{Chlorophyll } b = 20.13 \times A_{645} - 4.19 \times A_{662} \tag{2}$$

$$\text{Total chlorophyll} = 7.05 \times A_{662} + 18.09 \times A_{645} \tag{3}$$

$$\text{Carotenoid} = (1000 \times A_{470} - 1.90 \text{ chlorophyll } a - 63.14 \text{ chlorophyll } b)/214 \tag{4}$$

The Folin–Ciocalteu reagent found total phenol content (TPC) as per the procedure described in Horwitz [15]. Additionally, the flavonoids content (FC) of the extracts was measured by the method of Jia et al. [16]. The antioxidant activity was measured by converting the records of absorption of the samples into DPPH free radical scavenging percent using the following equation [17]

$$\text{Free radical scavenging percent} = \frac{\text{Sample absorption } - \text{ control absorption}}{\text{Control absorption}} \times 100 \tag{5}$$

To check the absorption of elements, we used a wet digestion extract. Then, the concentration of the elements was measured in mg/L and mg/kg with an atomic absorption device [18,19]. The hot extraction method AOAC Official Method 972.28 (41.1.22) determined seed oil content using a Soxhlet extractor (Buchi, B-811, Switzerland) [5]. After ensuring the normality of data, their combined analysis was performed in the SPSS software package. Additionally, the means were compared by Duncan's multiple range test at the $p < 0.05$ level.

### 3. Results and Discussion

*3.1. Thousand-Seed Weight*

The highest 1000 seed weight of 15.03 g was obtained from the plants treated with *Azotobacter* + chemical fertilizer, significantly different from the other treatments. The lowest one (13.73 g) was related to the control treatment. Additionally, the lonely application of *Azotobacter* and the control had a similar impact on 1000 seed weight (Table 2). Previous research findings indicated that biological fertilizers, especially in water deficit conditions, improved root growth and photosynthetic assimilation by expanding leaf area and enhancing pre-flowering photosynthesizing capacity, thereby improving 1000 seed weight by increasing post-flowering remobilization this photosynthesis from the source to the sink [20]. A study of the effect of different nutritional systems and biological fer-

tilizers on sunflowers showed that the highest 1000 seed weight was obtained from the mixed treatment of manure and chemical fertilizers [21]. According to Alami-Milani et al. [22], the highest 1000 seed weight was obtained from the application of 50% urea and superphosphate + Nitroxin and bio-super phosphate (a set of phosphate-dissolving bacteria). Ghasemi et al. [23] reported that the 1000 grain weight of maize was influenced by fertilizer type.

**Table 2.** Means comparison for the quantitative and qualitative traits of *Syrian cephalaria* as influenced by organic and chemical fertilizers.

| Treatment | 1000 Seed wt | Seed Yield per Plant | Biological Yield per Plant | Harvest Index | Oil % | Oil Yield per Plant (g) | Chl. *a* | Chl. *b* | Total Chl. | TPC | FC | DPPH |
|---|---|---|---|---|---|---|---|---|---|---|---|---|
| | (g) | | | | | | (mg/g FW) | | | | | |
| Control | 13.73 [c] | 6.99 [c] | 16.86 [e] | 41.44 [bc] | 24.09 [b] | 1.68 [c] | 1.63 [bc] | 1.47 [d] | 3.10 [c] | 26.55 [b] | 0.52 [a] | 51.62 [b] |
| *Azotobacter* | 13.76 [bc] | 6.99 [c] | 16.45 [e] | 42.30 [b] | 23.70 [b] | 1.66 [c] | 1.59 [c] | 1.58 [c] | 3.17 [c] | 26.96 [ab] | 0.55 [a] | 52.45 [ab] |
| *Azotobacter* + manure | 14.23 [bc] | 7.98 [bc] | 20.77 [c] | 38.54 [d] | 21.76 [c] | 1.74 [c] | 1.95 [a] | 1.72 [b] | 3.67 [ab] | 27.89 [a] | 0.56 [a] | 53.16 [a] |
| *Azotobacter*+ vermicompost | 14.53 [ab] | 9.55 [a] | 22.82 [b] | 41.85 [b] | 25.23 [a] | 2.41 [a] | 2.06 [a] | 1.96 [a] | 4.02 [a] | 22.49 [c] | 0.51 [a] | 49.20 [c] |
| *Azotobacter* + Phosphate Barvar II | 14.02 [bc] | 8.68 [ab] | 18.54 [d] | 46.77 [a] | 22.57 [c] | 1.96 [bc] | 1.77 [b] | 1.69 [b] | 3.46 [bc] | 25.87 [b] | 0.53 [a] | 51.28 [b] |
| *Azotobacter* + chemical fertilizer | 15.03 [a] | 9.97 [a] | 24.57 [a] | 40.30 [c] | 22.24 [c] | 2.22 [ab] | 2.04 [a] | 1.96 [a] | 4.00 [a] | 22.06 [c] | 0.42 [b] | 48.69 [c] |

TPC—Total phenol content; FC—Flavonoids content; Means with one similar letter did not show significant differences at $p < 0.05$ level.

### 3.2. Seed Yield

Seed yield was significantly affected by the fertilization treatments. Plants treated with *Azotobacter* + chemical fertilizer produced the highest seed yield of 9.97 g/plant compared to 6.99 g/plant seed yield in the control plants (no fertilization) and plants treated with *Azotobacter* (Table 2). Thus, the biological and organic fertilizers can optimally meet plant needs, and various nutritional sources may affect the release of nutrients and their preparation. So, biological fertilizers are essential for continuous plant nutrients availability [24]. Researchers have reported the higher yield of the integrated system by using organic fertilizers [25]. A study on the effect of integrated application of various fertilizer sources showed that *cephalaria* plants produced a higher seed yield when fed with a mixture of organic, biological, and chemical fertilizers than mycorrhiza treatments. Thus, the integrated nutritional system could affect all yield-influencing traits [26]. Similarly, we found that the plants treated with organic, biological, and chemical fertilizers outperformed the control plants and those treated only with *Azotobacter*. Therefore, the fact that the integrated use of fertilizers is influential on all traits that underpin yield.

### 3.3. Biological Yield

The highest biological yield (24.57 g/plant) was related to the integrated treatment of *Azotobacter* + chemical fertilizer treatment so that the integrated use of fertilizer sources had higher biological yield compared to *Azotobacter* treatment and control. The lowest biological yield (16.45 g/plant) was related to the *Azotobacter* treatment (Table 2). Awad et al. [27] reported that the integrated application of biofertilizer Nitorxin with chemical N fertilizer increased the biological yield of anise plants and remarkably reduced the N fertilizer rate. Concerning the effect of integrated use of organic, biological, and chemical fertilizers on the biological yield of *cephalaria*, it can be said that the integrated fertilization system improves organic content of soil, and then, by affecting moisture and nutrient uptake, storage, and availability [24,28], it increases yield components including the number of auxiliary branches and improves biological yield. The positive effects of vermicompost and manure may be associated with a higher organic content of the soil and a more balanced availability of macro and micronutrients in the soil, which can directly affect both vegetative and reproductive growth *cephalaria*. Despite the influential role of biological fertilizers in plant yields, the efficiency of N-fixing species and mycorrhiza can be considerably dictated by the availability of organic matter in their growth media [5].

### 3.4. Harvest Index

Harvest index is the ratio of economic yield (seed yield) to biological yield (dry weight of aerial parts of the plant) and reflects how assimilates are allocated within the vegetative parts of plants and flowers. The highest harvest index was 46.77%, obtained from the plants treated with *Azotobacter* + phosphate Barvar II. The integrated treatment of *Azotobacter* with organic, biological, and chemical fertilizers had a similar effect on the seed harvest index of *cephalaria*. The lowest harvest index (38.54%) was related to the *Azotobacter* + manure treatment (Table 2). The variation of harvest index depends on the variation of seed yield. A high harvest index is acceptable when it results from the overall increase in dry matter produced in the field or the increase in the share of the economic yield, or both of them. The availability of nutrients and water at the seed filling period enhances the harvest index, because their availability positively affects the current photosynthesis. Thus, nutrient deficiencies in the growth medium of a plant can influence photosynthetic allocation to different parts of the plant, which is mainly manifested as the allocation of less photosynthates to reproductive parts resulting in further loss of seed yield versus biological yield and, finally, the decline of harvest index. The critical factor in determining harvest index is the plant's responses to resource limitations [29]. Exposure to growth limitations reduces available resources such as radiation, water, and nutrients, and subsequently, plants mobilize more photosynthates to their underground parts. The plant's seed yield and biological yields are declined during the reproductive growth stage, and this decline mostly happens in seed number and weight.

### 3.5. Seed Oil Content

The results of ANOVA showed that oil content was significantly ($p < 0.05$) affected by the fertilization treatments (Table 3). The highest oil content of 25.23% was related to the *Azotobacter* + vermicompost treatment, so that the integrated use of *Azotobacter* with organic, biological, and chemical fertilizers had a similar impact on the oil content of *cephalaria* seeds. The lowest oil content of the seeds was 21.76% observed in the treatment of *Azotobacter*+ manure (Table 2). The integrated application of *Azotobacter* and *Azospirillum* + N fertilizer treatments could improve soil physicochemical characteristics, thereby increasing nutrient uptake, $CO_2$ absorption, and photosynthesis in canola plants, resulting in the improvement of their oil content [30]. It has been reported that, when more N fertilizers are used, more nitrogen-containing precursors are formed, increasing the synthesis of proteins, and consequently, less matter would be available to be converted to oil. However, when an integrated fertilization system is used, a balance is made between protein synthesis and oil formation in plants [31]. During the seed filling period, proteins are first synthesized and then, oil is formed. As the seed filling period is prolonged, more oil is formed, and proteins are converted to oil even in the final stages. However, when this period is short, there will not be enough chance for oil formation, and since proteins are formed at the early stages, they will dominate. On the other hand, when more oil is contained in seeds, their proteins decline because seed size is constant [32].

**Table 3.** Analysis of variance for the quantitative and qualitative traits of *Syrian cephalaria* as influenced by organic and chemical fertilizers.

| S.O.V. | df | 1000-Seed Weight | Seed Yield/Plant | Yield/Plant | HI | Oil (%) | Oil Yield/Plant (g) | Chl. *a* | Chl. *b* | Total Chl. ll | TPC | FC | DPPH | Ca (mg/kg) | Fe | Cu | Zn |
|---|---|---|---|---|---|---|---|---|---|---|---|---|---|---|---|---|---|
| Treatment | 5 | 0.746 | 4.79 | 32.40 | 7.01 | 2.33 | 0.293 | 0.138 | 0.117 | 0.471 | 17.837 | 0.007 | 9.496 | 11.191 | 181.23 | 26.13 | 22.07 |
| Experimental error | 12 | 0.587 ns | 0.565 ** | 0.526 ** | 0.156 ** | 0.094 ** | 0.036 ** | 0.008 ** | 0.003 ** | 0.053 ** | 0.342 ** | 0.002 ** | 0.582 ** | 0.653 ** | 6.64 ** | 0.454 ** | 0.244 ** |
| C.V. (%) | | 5.39 | 8.99 | 3.62 | 0.94 | 1.31 | 9.78 | 4.86 | 3.16 | 6.44 | 2.31 | 8.76 | 1.49 | 6.38 | 1.53 | 2.42 | 1.16 |

TPC—Total phenol content; FC—Flavonoids content; ns and ** show insignificance and significance at $p < 0.01$, respectively.

### 3.6. Oil Yield per Plant

According to the comparison of the mean, the integrated application of *Azotobacter* and vermicompost produced the highest oil yield per plant (2.41 g/plant), and it significantly differed from that of the integrated *Azotobacter* + manure and *Azotobacter* + chemical fertilizer treatments. Plants fertilized with *Azotobacter* alone had the lowest oil yield per plant of 1.66 g (Table 2). Since oil yield is grain yield multiplied by oil percentage, the significant difference in the oil yield may be associated with the significant difference in the grain yield of different fertilization treatments. The integrated application of organic, biological, and chemical fertilizers can be directly beneficial for plant growth by increasing the uptake of N, the synthesis of phytohormones, and the dissolution of minerals [30,33]. As a result, it increases seed yield, and finally, oil yield will also increase. Oil yield is the main target of growing and developing oilseeds such as *cephalaria*. It seems that, presently, the most reasonable way to achieve a high oil extraction rate per unit area is to increase seed yield due to the low range of oil percentage variations by environmental and nutritional factors.

### 3.7. Photosynthesizing Pigments

The highest chlorophyll content (2.06 mg/g FW) was obtained from *Azotobacter* + vermicompost treatments. However, it did not differ from *Azotobacter* + manure and *Azotobacter* + chemical fertilizer treatments, which had the same impact on chlorophyll content (Table 2). The application of *Azotobacter* + vermicompost treatment resulted in the highest contents of chlorophyll *b* (1.96 mg/g FW) and total chlorophyll (4.02 mg/g FW), but it did not differ from the integrated use of *Azotobacter* and chemical fertilizer treatment significantly (Table 2). The plants treated with *Azotobacter* treatment had the lowest content of chlorophyll *a* (1.59 mg/g FW), chlorophyll *b* (1.58 mg/g FW), and total chlorophyll (3.17 mg/g FW) (Table 2). The significant increase in some fertilizer treatments versus control implies the significant effect of integrated use of biological, organic, and chemical fertilizers on photosynthesizing pigments. It appears that when organic, biological, and chemical fertilizers are applied together, not only is the N requirement of the plants supplied, but N wastage (leaching, sublimation, or fixation processes) is also reduced. Then, due to the process of mineralization, N is gradually made available to plants in absorbable forms, which increases their vegetative growth during the growing season, resulting in their high chlorophyll content in the integrated fertilization system. It has been reported that the application of biological and chemical fertilizers increased the N content of the plants, thereby increasing their chlorophyll contents and, subsequently, their ability to absorb sunlight and produce photosynthates, which resulted in their higher growth and yield [34]. In this study, it seems that the application of organic and biological fertilizers enhanced leaf chlorophyll synthesis and content by hindering N leaching and its more supply, contributing to the synthesis of growth stimulators, increasing soil microbial population and escalating the availability of nutrients and their more efficient uptake [35]. Thus, the plants treated with these fertilizers exhibited a higher rate of N uptake, and consequently, their chlorophyll content was improved due to its direct relationship with N content.

### 3.8. Total Phenol Content and Flavonoids Content

The highest total phenol content (27.89 mg gallic acid/g DM) was obtained from *Azotobacter* + manure treatment. However, the lowest TPC (22.06 mg gallic acid per g DM) was related to *Azotobacter* + chemical fertilizers treatment ($p \leq 0.05$, Table 2). Phenol compounds are usually influenced by genetic factors and environmental conditions, including nutrition [36]. The difference in phenol content can affect the antioxidant activity of plants, because most phenol compounds are a good source of natural antioxidants in plants. It has been documented that the phenol content of fennels treated with 50% chemical fertilizer + 50% organic fertilizer + biological fertilizer was higher than that of (unfertilized) control [37]. There is a report that applying chemical fertilizer increased the total phenol

content of savory [38]. Given the hypothesis of the balance between carbon and minerals and the hypothesis of the growth differentiation, there is a mutual relationship between primary and secondary metabolisms [39], so the increase in nutrients of manure-treated soils increases the net photosynthesis rate of the plant and results in the increasing of the activity of enzymes involved in starch and protein biosynthesis in the synthesis of secondary compounds [40]. There is a direct relationship between the increase in phenol compounds and the increase in plant carbohydrates. Since carbohydrates are the skeleton required for the synthesis of phenol compounds, their increase means more substrate for phenol compounds, which may be associated with the allocation of more carbons to the shikimic acid pathway [41].

The means comparison showed that the highest flavonoids content (FC) (0.56 mg quercetin/g DM) was obtained from the application of *Azotobacter* + manure treatment. The lowest one (0.42 mg quercetin/g DM) was related to control and *Azotobacter* + chemical fertilizer treatment (Table 2). Flavonoids have antioxidant properties and are involved in regulating enzymatic activities and the synthesis of primary metabolites. The amount of flavonoids in different plant species depends on growth stage, tissue, cultivar, environmental stresses such as UV stress and drought, soil conditions, tillage, pests and diseases, and fertilizer application [42]. Meanwhile, flavonoids inhibit oxidative stress by directly engaging in reduction reactions and indirectly chelating iron [43]. Research has shown that the application of organic, biological, and chemical fertilizers had a stimulating effect on the accumulation of flavonoids in broccoli [44]. It is well established that higher concentrations of flavonoids and phenols can be identified by the role of organic fertilizers in biosynthesis, which creates a pathway for acetate shikimate and increases the synthesis of flavonoids and phenols [45].

### 3.9. DPPH Antioxidant Activity

The means the comparison showed that the highest DPPH antioxidant activity (53.16%) was related to the application of *Azotobacter* + manure treatment, whereas the lowest one (48.69%) was observed in the treatment of *Azotobacter* + chemical fertilizer treatment (Table 2). It seems that the simultaneous application of mycorrhiza + manure treatment increased the antioxidant activity of *cephalaria* seeds by improving the physical and chemical properties of the soil and the gradual release of nutrients to plants. It has been reported that the application of organic fertilizers effectively enhances the antioxidant activity of fennels by positively influencing the physical and chemical properties of soil, increasing soil organic matter, and improving plant access to more nutrients [26]. Thus, it seems that the increased antioxidant activity of thyme plants treated with organic fertilizer was related to the improvement of soil physical and chemical properties, the gradual release of nutrients, and the increased availability of N and P to the plants. Ghasemzadeh et al. [46] reported that the antioxidant capacity of the plants was increased by the increase in total phenol and flavonoid contents during the application of organic fertilizers.

### 3.10. Concentrations of Trace Elements

According to the comparison of the mean, the highest Ca content (14.78 mg/kg) was observed in the plants treated with *Azotobacter* + vermicompost treatment and the lowest (9.44 mg/kg) from the control treatment, so that the integrated fertilization treatments had a significant difference from the control treatment. Furthermore, all applied fertilization treatments increased Ca content versus control significantly (Figure 1).

Concerning Fe content, the highest value (174.70 mg/kg) was related to the *Azotobacter* + vermicompost treatment and the lowest (153.69 mg/kg) to the control treatment, and these two extreme values differed significantly. Additionally, all fertilization treatments resulted in a significantly higher Fe content than in the control (Figure 1). The highest Cu content (31.25 mg/kg) was related to the *Azotobacter* + vermicompost treatment, not differing significantly from integrated fertilizer use. The lowest (22.98 mg/kg) was obtained from the control plants (Figure 1). The means comparison indicated that the application of

*Azotobacter* + vermicompost treatment was related to the highest Zn content (44.82 mg/kg) and the lowest (37.21 mg/kg) to the control plants (Figure 1). The *Azotobacter* + vermicompost treatment did not differ significantly from the *Azotobacter* + manure and *Azotobacter* + chemical fertilizers treatments, and the two latest treatments had a similar impact on Zn content. By their metabolic activities, microorganisms convert minerals and organic matter of soils from one form to another and change the availability of the nutrients for plants and other organisms of the soils. The nutrient cycle and soil formation play an essential role in a decomposing organic matter [47]. The higher concentration of Fe in the plants treated with mycorrhiza + vermicompost may be related to the capability of vermicompost microorganisms and mycorrhiza in producing siderophore.

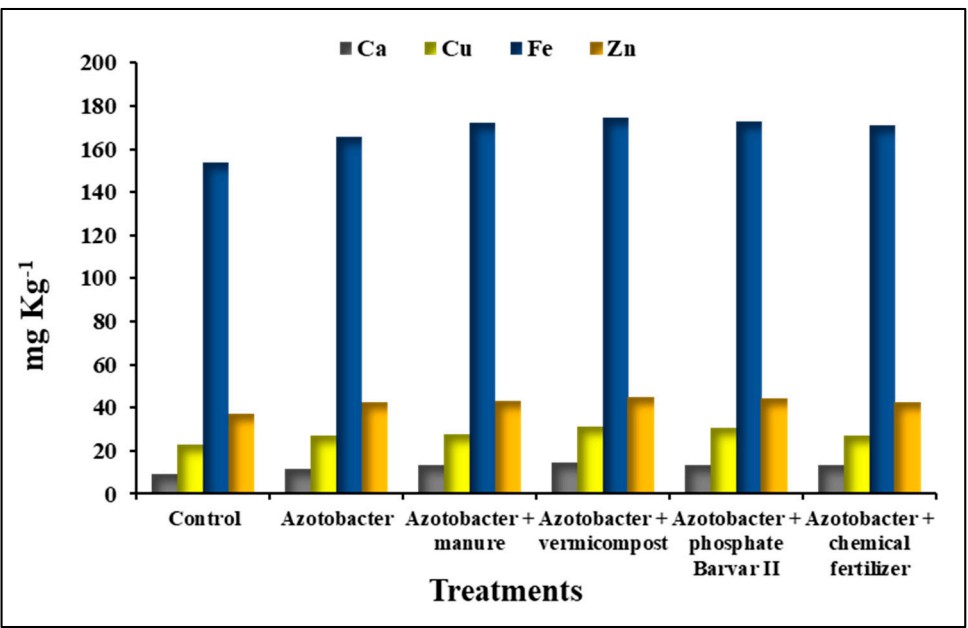

**Figure 1.** The concentration of Ca, Fe, Cu, and Zn, influenced by the applied fertilization treatments.

On the other hand, vermicompost itself provides the plants with nutrients. These are the possible reasons for the higher Fe content in plants exposed to mycorrhiza + vermicompost treatments.

Siderophores are reportedly organic compounds with a low molecular weight that have a strong tendency to form complexes with some cations, including Fe. Plants can use siderophores secreted by bacteria to supply their required Fe [48]. It has been established that mycorrhiza chelated Fe by secreting various siderophores and thereby increased its uptake and mobilization in peanut and sorghum plants [49]. Meanwhile, the simultaneous application of some biological fertilizers improves the use of nutrients owing to their synergistic relationship [35]. Egamberdiyevva and Hoflich [50] found that *Azospirillum* had a significant effect on increasing Cu content of the seeds, mainly emanating from the synthesis of growth regulators by the bacteria and their impact on root growth and the uptake of water and nutrients from the soil. Vermicompost contains beneficial aerobic microorganisms such as *Azotobacter* and is free from anaerobic bacteria, fungi, and pathogens. Compared to the initial maternal source, vermicompost has lower dissolved salts and higher cation exchange capacity and humic acid.

On the other hand, vermicompost contains nutrients in forms that plants can readily absorb [51]. There is evidence that manure application increased the Fe and Zn content of barley plants [52]. Research suggests that the integrated use of biological and organic fertilizers effectively increases the activity of acid phosphatase and alkaline phosphatase around roots resulting in an enhanced level of P in soils and an escalated rate of N, Zn, Cu, and Fe uptake [53,54]. The increased uptake of nutrients associated with the inoculation with growth-promoting bacteria can be attributed to the morphological changes in plant

roots, significantly increasing the number, length, and thickness of the roots, as well as to the higher cation exchange capacity of the inoculated plants [19,34,55].

## 4. Conclusions

The results of ANOVA showed that most studied traits (seed yield per plant, biological yield harvest index, oil content, oil yield per plant, photosynthesizing pigments, total phenol content and flavonoids content, antioxidant activity, and concentration of trace elements) were significantly ($p \leq 0.01$) influenced by different fertilization sources. On the other hand, fertilization treatments did not influence 1000 seed weight (Table 3).

The results revealed that the simultaneous application of *Azotobacter* + vermicompost treatment performed the best among all treatments in improving the quantitative and qualitative traits of *Syrian cephalaria*. The optimal levels of the traits about antioxidant properties were obtained from the integrated application of the fertilizers. The measured concentrations of elements in plant tissues were the highest in *Azotobacter* + vermicompost treatment. It is inferred from the results that the integrated application of organic, biological, and chemical fertilizer is an effective way to improve soil fertility, nutrient uptake, and thereby, many quantitative and qualitative traits of *cephalaria*. Given the current tendency towards reducing the use of chemical fertilizers to alleviate the contamination of ground tables and crops, increase production efficiency, and achieve sustainable agriculture goals, it is recommended to use organic, biological, and chemical fertilizers within an integrated system.

**Author Contributions:** Writing—Original draft, A.R.; Conceptualization, Supervision, R.A.; Formal analysis; S.S.M., Writing—Review and Editing, H.A.E.E., S.Z.H. and R.Z.S. All authors have read and agreed to the published version of the manuscript.

**Funding:** This work was funded by RMC, Universiti Teknologi Malaysia, and financial support through grant No. R.J130000.7344.4C240 and R.J130000.7609.4C336 and R.J130000.7344.4B200.

**Institutional Review Board Statement:** Not Applicable.

**Informed Consent Statement:** Not Applicable.

**Data Availability Statement:** All the data is available in the manuscript file.

**Acknowledgments:** The authors thank RMC, Universiti Teknologi Malaysia, for financial support.

**Conflicts of Interest:** The authors declare that they have no conflict of interest.

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
