# Peer review of "Effect of Different Biological and Organic Fertilizer Sources on the Quantitative and Qualitative Traits of Cephalaria syriaca"

_horticulturae, doi:10.3390/horticulturae7100397_

Round 1

Reviewer 1 Report

The work has a very good, future-oriented approach. The results regarding the yields and ingredients are all plausible and conclusive. The work could be a "milestone" for biologically oriented plant nutrition.

However, it has very serious methodological shortcomings. Line 87/88 "The experimental treatments are including (1) control, (2) Azotobacter, (3) Azotobacter + manure, (4) Azotobacter + vermicompost, (5) Azotobacter + phosphate Barvar II, and (6) Azotobacter + chemical fertilizer.

Why were no pure variants manure / vermicompost / phosphate Barvar II, and chemical fertilizer included in the investigation? How was the quantification of the additives done (manure / vermicompost / phosphate Barvar II, and chemical fertilizer). What amounts were fertilized and what was the nutrient content of the additives?

The effect of the additives in combination with Azotobacter may not only result from different nutrient applications by manure / vermicompost / phosphate Barvar II, and chemical fertilizer. The amount and content must be supplemented. With the additive phosphate Barvar II, an internationally insignificant fertilizer, I only found out through research that it already contains bacteria and mycorrhizal fungi. These gross deficits must be worked on before publication.

The missing variants are certainly not to be added.

It would certainly be desirable to also consider very current literature! e.g. Eulenstein, F., Schindler, U., Ahlborn, J., Scharschmidt, P., Saljnikov, E., Behrendt, A. (2021) Influence of the application of mycorrhizal fungi and Bacillus amyloliquefaciens on the yields of three vegetables and a grass with organic fertilization on peat-free growing media in organic plant production. In: Vandecasteele, B., Viaene, J. (eds), Proceedings of the II International Symposium on Growing Media, Soilless Cultivation, and Compost Utilization in Horticulture: Ghent, Belgium, August 22-27, 2021. International Society for Horticultural Science , Leuven, pp. 247-254. 

Author Response

Comments

Answer

Reviewer 1

However, it has very serious methodological shortcomings. Line 87/88 "The experimentaltreatments are including (1) control, (2) Azotobacter, (3) Azotobacter + manure, (4)Azotobacter + vermicompost, (5) Azotobacter + phosphate Barvar II, and (6) Azotobacter+ chemical fertilizer.

Why were no pure variants manure / vermicompost / phosphate Barvar II, and chemicalfertilizer included in the investigation?

Because in the presence of organic fertilizers, biofertilizers can be more efficient. Our aim was to use compound fertilizers and due to the limitations in experimental treatments, it was not possible to apply separate treatments.

How was the quantification of the additives done (manure / vermicompost / phosphate Barvar II, and chemical fertilizer).

What amounts were fertilized and what was the nutrient content of the additives?

The effect of the additives in combination with Azotobacter may not only result fromdifferent nutrient applications by manure / vermicompost / phosphate Barvar II, andchemical fertilizer.

The amount and content must be supplemented. With the additivephosphate Barvar II, an internationally insignificant fertilizer, I only found out throughresearch that it already contains bacteria and mycorrhizal fungi.

These gross deficits must be worked on before publication.

The missing variants are certainly not to be added

After performing the calculations, fertilizers were applied to the soil of the Cephalaria plant requirement, vermicompost (6.8 ton ha−1), manure (6.3 ton ha−1 ) and chemical fertilizer (Urea: 110 kg ha−1 + Triple superphosphate: 60 kg ha−1 + Potassium sulfate: 50 kg ha−1 + Micronutrients: 23 kg ha−1 )

Before sowing, the seeds were inoculated according to the procedure recommendation of Green Biotech Inc, with biological Azotobacter fertilizer and Phosphate Barvar II, which contained Azotobacter vinelandii (the strain O4) and strain P5 of Pantoea and strain P13 of Pseudomonas putida (concentration formulated product: 109 CFU/g)

Corrected=line 95

Reviewer 2 Report

This manuscript "Effect of different biological and organic fertilizers sources on the quantitative and qualitative traits of Cephalaria syriaca L." is fairly well written and worth publishing in sustainability but it needs revisions before it is published. The comments and suggestions are annotated in the manuscript. I strongly suggest to get a native English speaker to read the manuscript before resubmit. 

Author Response

Reviewer 2

Please check format

Corrected

Syrian Cephalaria or Syrian cephalaria  or Syrian cephalaria

Please make sure which one is correct  

Corrected= Done throughout the text

Syrian cephalaria (Cephalaria syriaca (L.)

Italic Azotobacter

Corrected= Done throughout the text

Some ar  e mention in difference way such as Cephalaria or Cephalaria or cephalaria. Please make sure and change in same style  

Corrected= Done throughout the text

cephalaria

Which one do you recommend the most for Cephalaria?  

organic and biological inputs

Corrected and replaced with “combined fertilizers including organic and biological inputs”

arrange in alphabetical order  

Keywords:

Corrected=line 36

Italic   Cephalaria

Corrected= Done throughout the text

between numerals please use en-dash, check throughout the manuscript  

Corrected= Done throughout the text

Italic  Cephalaria

Corrected= Done throughout the text

Meanwhile, the Cephalaria has a good oil content (21-26%), so it is sometimes considered an oilseed

why sometimes?, please can you clarify this sentence  (

Explained in the text. The oil contains 7.8% epoxy acid, calculated as epoxy oleic acid, which makes its use as an edible oil rather difficult. Epoxy fatty acids have high-value uses in glues, resins, and surface coatings But renders it usable in industries using epoxidized oils. Due to its high content of myristic acid, the oil is very suitable for soapmaking as well.

L. Only mention first time is enough

Syrian Cephalaria L.  

Corrected=line  81

Italic   Cephalaria

Corrected= Done throughout the text

Syrian Cephalaria  

Corrected= Done throughout the text

Syrian cephalaria

What is the treatment for control? Normal seeds without inoculated Azotobacter?

Corrected=line 91

(no fertilization)

Before sowing, the seeds were inoculated with biological Azotobacter fertilizer,

Control as well?  

Control was without inoculation.

 (1)??

Corrected=line 120

You have mentioned only higher and lower ones, how about other treatments?  

The graphs are self-explanatory. Moreover, we mentioned in the text” Also, the lonely application of Azotobacter and the control had a similar impact on 1000-seed weight”

(no fertilizer application). This need to mention in M&M  

Corrected=line 91

If you want to find information to support your previous sentences please make it clear and make sure they are similar with your results. This is difficult for the reader to follow with your results.  

Corrected=line 144

please be consistent

Syrian cephalaria

Corrected= Done throughout the text

Syrian cephalaria

DPPH stands for?

Yes: (2,2-diphenyl-1-picrylhyydrazyl)

Can you move exponent near number

Corrected=line 156

superscript

Corrected=line 157

Did you use mycorrhiza treatments? Which one?  

No, we did not use mycorrhiza, and here we mean the superiority of compound fertilizer, which also included biofertilizer.

The highest biological yield (24.57 g/plant) was related to the integrated treatment of Azotobacter + chemical fertilizer treatment so that the integrated use of all fertilizer sources had a similar effect on the biological yield of Cephalaria.

Treatments show are significant? I don't think they are similar   

Corrected

This should be conclusion  

Corrected=line 379

make sure which style is correct   Cephalaria

Corrected= Done throughout the text

Round 2

Reviewer 2 Report

Dear authors,

This manuscript "Effect of different biological and organic fertilizer sources on the quantitative and qualitative traits of Cephalaria syriaca" can be accepted after the suggested is done.
Please see the attached file with few annotated comments.

Kind regards
